Modeling the bidirectional glutamine/ammonium conversion between cancer cells and cancer-associated fibroblasts

http://orcid.org/0000-0001-5875-1470 Hinow Peter 1 hinow@uwm.edu
Pinter Gabriella 1
Yan Wei 2
Wang Shizhen Emily 2
1 Department of Mathematical Sciences, University of Wisconsin-Milwaukee , Milwaukee, WI , USA
2 Department of Pathology, University of California, San Diego , La Jolla, CA , USA
Silva Pedro
Electronic publication date: 2021 Jan 13
Publication date: 2021
Volume: 9
Electronic Location ID: e10648
Received 2020 Sep 10; Accepted 2020 Dec 4
Copyright: © 2021 Hinow et al.
Copyright year: 2021
Copyright holder: Hinow et al.
License: This is an open access article distributed under the terms of the Creative Commons Attribution License, which permits unrestricted use, distribution, reproduction and adaptation in any medium and for any purpose provided that it is properly attributed. For attribution, the original author(s), title, publication source (PeerJ) and either DOI or URL of the article must be cited.
License URL: https://creativecommons.org/licenses/by/4.0/

Keywords: Cancer-associated fibroblasts, Glutamine/ammonium metabolism, Mathematical modeling

Funding: Simons Foundation 278436 National Institutes of Health (NIH) R01CA218140 Peter Hinow’s visit to UCSD was made possible with support from the Simons Foundation grant “Collaboration on Mathematical Biology” (Award 278436 to Peter Hinow). This work is supported by the National Institutes of Health (NIH) grant R01CA218140 to Shizhen Emily Wang. There was no additional external funding received for this study. The funders had no role in study design, data collection and analysis, decision to publish, or preparation of the manuscript.

==============================
Like in an ecosystem, cancer and other cells residing in the tumor microenvironment engage in various modes of interactions to buffer the negative effects of environmental changes. One such change is the consumption of common nutrients (such as glutamine/Gln) and the consequent accumulation of toxic metabolic byproducts (such as ammonium/NH4). Ammonium is a waste product of cellular metabolism whose accumulation causes cell stress. In tumors, it is known that it can be recycled into nutrients by cancer associated fibroblasts (CAFs). Here we present monoculture and coculture growth of cancer cells and CAFs on different substrates: glutamine and ammonium. We propose a mathematical model to aid our understanding. We find that cancer cells are able to survive on ammonium and recycle it to glutamine for limited periods of time. CAFs are able to even grow on ammonium. In coculture, the presence of CAFs results in an improved survival of cancer cells compared to their monoculture when exposed to ammonium. Interestingly, the ratio between the two cell populations is maintained under various concentrations of NH4, suggesting the ability of the mixed cell system to survive temporary metabolic stress and sustain the size and cell composition as a stable entity.

Introduction

Recent years have seen an increased appreciation of cancer as an ecological problem (Nagy, 2005; Kim et al., 2010). This has also led to a large number of sophisticated mathematical models, see Enderling & Chaplain (2014), Altrock, Liu & Michor (2015), Kuang, Nagy & Eikenberry (2016) for some contemporary introductions to mathematical modeling of cancer. Solid tumors are not merely masses of cancer cells, but are populated by endothelial cells, immune cells and fibroblasts. Cancer-Associated Fibroblasts (CAFs), also referred to as myofibroblasts, are the major cellular component of tumor stroma. CAFs may include heterogeneous subpopulations originating from normal fibroblasts upon activation by cancer-derived stimuli (Mueller & Fusenig, 2004; Kalluri & Zeisberg, 2006; Kojima et al., 2010), cancer or normal epithelial cells undergoing epithelial-to-mesenchymal transition (Petersen et al., 2003; Kalluri & Neilson, 2003), or—as more recently proposed—bone marrow-derived mesenchymal stem cells (Karnoub et al., 2007; Quante et al., 2011). See Fig. 1A for a conceptual model.

Figure 1 (A) A mechanism for reprograming of normal fibroblasts (FBs) into cancer-associated fibroblasts (CAFs) by cancer cells (CC) through secreted effectors.

These include cytokines and extracellular vesicles. The dashed arrow indicates a potential therapeutic intervention to normalize CAFs and the tumor microenvironment. The normal fibroblasts would no longer aid the cancer cells in their survival in adverse metabolic conditions. See also Fig. 11. (B) A conceptual model for the bidirectional Gln/NH4+ metabolism of both cancer cells and CAFs in monoculture. Both cell types are able to convert waste into nutrient, however the cancer cells decline in number when they have to do that. The combined populations show an improved temporary survival of cancer cells compared to the cancer cell monoculture.

Various coculture experiments with CAFs have shown that their properties are markedly different from their counterparts in healthy tissue. For example, fibroblasts from human prostate cancers are able to promote proliferation and initiate a pathway to malignancy in epithelial cells from benign prostate hyperplasia (Bhowmick, Neilson & Moses, 2004). It is well accepted that CAFs promote tumor growth and progression by releasing growth factors and cytokines, as well as components and modifiers of extracellular matrix into the tumor milieu (Mueller & Fusenig, 2004; Kalluri & Zeisberg, 2006). A better understanding of the relationship between a tumor and the tumor microenvironment is needed for novel therapeutic approaches (Kim et al., 2010).

Cancer cells as well as non-cancer cells residing in the same tumor milieu share all extracellular metabolites including essential nutrients (e.g., glucose and amino acids) and toxic wastes (e.g., lactic acid and ammonium). As a result of oncogenic signaling, cancer cells often exhibit increased uptake of key nutrients such as glucose and glutamine to fuel rapid growth and proliferation. As a solid tumor grows, it often suffers from a lack of blood supply because of insufficient or immature tumor-associated blood vessels. Thus many tumor cells experience metabolic stress, including deprivation of nutrients and accumulation of metabolic wastes. Cancer’s exploitation of nutrients and the generation of cancer-derived metabolic byproducts also influence stromal cells residing in the tumor milieu. Recent studies have revealed interesting patterns of metabolic interaction between cancer and stromal cells (Pavlides et al., 2009; Lisanti, Martinez-Outschoorn & Sotgia, 2013; Martins et al., 2013; Fong et al., 2015; Loo et al., 2015; Zhang et al., 2015; Zhao et al., 2016). Our group reported that breast cancer cells, through secreting extracellular vesicles, educate CAFs to convert cancer-produced lactic acid and ammonium into pyruvate and glutamine, respectively, which are subsequently used to fuel cancer cells while detoxifying the metabolic environment of tumor by removing metabolic wastes (Yan et al., 2018). Certain cancer cells can also partially recycle ammonium into central amino acid metabolism (Spinelli et al., 2017). However, because cancer cells and CAFs may exhibit different survival and proliferation capabilities under varying levels of glutamine and ammonium, it is of significance to compare the growth of tumor mass (total cell number) in the presence or absence of CAFs to understand the exact contribution of CAFs.

The goal of this article is to construct and parametrize a mathematical model for the symbiotic and synergistic interactions between a population of cancer cells and a population of stromal fibroblasts. The model is complemented by in vitro experimental observations in which either cell type is grown by itself (monoculture) and in which the two cell types are seeded in a 1:1 mixture (coculture). We strive to keep the mathematical complexity at a minimum, but to allow enough flexibility to expand the model in future work. As an example of the nutrient-waste dynamics and metabolic interaction between cancer cells and CAFs, we chose to model the recently reported ability of cancer cells and cancer-reprogrammed CAFs to convert ammonium (NH4+), an end product and toxic waste generated by glutamine catabolism, back to glutamine (Gln) to nourish the tumor environment. The mathematical model is formulated in terms of ordinary differential equations. It allows to focus on the specific forms of the interaction and competition terms.

The conceptual model is described in Fig. 1B. Spinelli et al. (2017) reported recently that cancer cells are able to convert NH4+ to Gln. However, this comes at a price, namely the decline of the population size in our experimental model system of cultured breast cancer cells (MDA-MB-231). We first create a mathematical model for a single cell type, either CAFs or cancer cells. The model contains the same terms, but results in different parameter values for the two monoculture scenarios. Later we merge the two models for the coculture scenario to investigate the behavior of the two cell types when they co-reside in the same metabolic environment. As it will turn out, the cell population data will require an extension of the simple merged coculture model. We introduce an extra “stress factor” that results in increased cell death after 48 h.

Methods

Cells

The MDA-MB-231 human breast cancer cell line was obtained from American Type Culture Collection (Manassas, VA, USA) and maintained in the recommended medium. Patient-derived primary fibroblasts CAF265922 (denoted as CAF in the following) were isolated from a triple-negative breast tumor and maintained in Iscove’s Modified Dulbecco’s Medium (Thermo Fisher Scientific, Waltham, MA, USA) supplemented with 20% fetal bovine serum as previously described (Tsuyada et al., 2012). All cell culture experiments used dialyzed fetal bovine serum (10,000 MW cutoff) to minimize the influence of serum-derived small molecules including amino acids and salts. Cells were tested to be free of mycoplasma contamination and authenticated by using the short tandem repeat profiling method. Purity of CAFs was ensured by fluorescence activated cell sorting using PDGFRβ as a marker. CAFs in culture were frequently checked to confirm they were negative for EpCAM or CD31 and positive for PDGFRβ and Vimentin. For all experiments and before cell seeding, CAFs were pretreated for 24 h with extracellular vesicles (EVs) collected from MDA-MB-231 cells at 2 μg of EVs (equivalent to those collected from 5 × 106 producer cells) per 2 × 105 recipient cells as described in Yan et al. (2018) to simulate education by cancer-secreted EVs.

Measurement of cell number

For single cell type culture, MDA-MB-231 cells or CAFs were seeded at 8 × 104 (time 0 h) per well on 6-well plates in Dulbecco’s Modified Eagle’s Medium (DMEM) without glucose, glutamine, and sodium pyruvate (Corning, Corning, NY, USA) that was supplemented with 3 g/L glucose, 10% fetal bovine serum and indicated levels of glutamine (0, 1, 2 or 4 mM) and ammonium chloride (NH4Cl; 0, 5, 10 or 25 mM). At indicated time points (24, 48 and 72 h), cells were stained with trypan blue (to label dead cells), and live cell numbers were determined using a TC20 automated cell counter (Bio-Rad Laboratories, Hercules, CA, USA). For coculture, MDA-MB-231 cells labeled with PKH67 green fluorescent cell linker (Sigma-Aldrich, St. Louis, MO, USA) and CAFs labeled with PKH26 red fluorescent cell linker (Sigma-Aldrich, St. Louis, MO, USA) were mixed at 1:1 ratio, and a total of 8 × 104 mixed cells were seeded. At indicated time points, numbers of each cell type were determined based on the different fluorescent labels.

Measurement of glutamine and ammonium

Cells were seeded and cultured as described above. At indicated time points (including time 0), the conditioned medium was collected, cleared by centrifugation, and subjected to measurements of glutamine and ammonium concentrations using a glutamine colorimetric assay kit (BioVision, Milpitas, CA, USA) and an EnzyChrom™ ammonia/ammonium assay kit (BioAssay Systems, Hayward, CA, USA), respectively. All experiments were done in triplicates and the averages were chosen for fitting the model parameters.

Model implementation

Numerical simulations of the ordinary differential equations were carried out using the function NDSolve of Mathematica (Wolfram Research, Champaign, IL, USA). The parameters were determined by fitting the model output to the experimental data using the function fminsearch of Matlab (MathWorks, Natick, MA, USA). The quadratic objective function is (1) F(θ)=∑k|u(tk;θ)−w(tk)|2,

where u(·;θ) denotes the full solution of the parameter-dependent ordinary differential equations and w are the experimental data. The sum includes all available data points for one cell type, whether treated with NH4+ or Gln. The minimization procedure uses the Nelder-Mead algorithm (Lagarias et al., 1998). Bootstrapping is used to re-sample the data set to create simulated data sets. For each such data set optimal parameters are found so that inferences could be drawn about specific parameters, in particular, the 95% confidence intervals could be calculated (Givens & Hoeting, 2013). The data is assumed to follow the model d = dψ + ε, where ε = (εCAF,εCC) and εCAF∼N(0,σCAF2) and εCC∼N(0,σCC2) are independent. Using the identified parameter set ψ^, the sample variances are estimated and used to generate random error terms to create simulated data sets. A total of 300 such data sets were utilized in the calculation of the 95% confidence interval for each parameter.

Results

CAFs grown alone

In the first monoculture scenario, the CAFs are grown with either Gln or NH4+ supplied at various concentrations, see Figs. 2 and 3. If the CAFs are presented with Gln, they convert it nearly entirely to NH4+, see the blue curves in Figs. 3A, 3C and 3E. This can be explained by the efficient glutaminolysis in the cells, generating NH4+ and glutamate. The latter can be further metabolized to α-ketoglutarate to enter the citric acid cycle for energy production. On the other hand, if CAFs are supplied with NH4+ they convert it to Gln, although not entirely, see the red curves in Figs. 2A, 2C and 2E. To make this possible, we introduce to the model a second source of chemical energy, which could be other metabolites, including amino acids such as glutamate, as well as glucose-derived α-ketoglutarate. The cells draw from this energy source to carry out the conversion of NH4+ to Gln, but also when they are supplied Gln. Note that R will be decreasing, as it is not replenished during the 72 h of observation in our experimental protocol. The CAFs can grow on both Gln and the other source and grow in the presence of NH4+ when this other source of energy exists.

Figure 2 (A, C and E) The concentrations of NH4+ and Gln when the CAFs are supplied with NH4+; (B, D and F) the number of live CAFs.

Here and in all following figures, solid lines represent the model simulations and dots represent the experimental data.

Figure 3 (A, C and E) The concentrations of NH4+ and Gln when the CAFs are supplied with Gln; (B, D and F) the number of live CAFs.

Let A(t), W(t), R(t) and X(t) denote the concentration of Gln, the concentration of NH4+, the chemical energy and the number of live CAFs at time t. Then we have (2a) dAdt=−k1AX+c2k2WRX,

(2b) dWdt=c1k1AX−k2WRx+hX,

(2c) dRdt=−k2WRX−k3RX,

(2d) dXdt=(r1AK1+A+r2RK2+R−d1W−d2)X.

In this model, k1 in Eqs. (2a) and (2b) denotes the rate of conversion of Gln to NH4+. The constant c1 is a dimensionless number that accounts for how much NH4+ is actually produced from Gln, whereas h includes the production of NH4+ from other possible sources. Similarly, k2 denotes the rate of conversion of NH4+ to Gln, while c2 is a dimensionless number that accounts for how much Gln is actually produced from NH4+. Both processes are proportional to the number of cells present, X. The conversion of NH4+ also requires the alternative energy source R to be present, which gets depleted as a result. The alternative energy is consumed by the cells at a rate k3 in Eq. (2c), independent of the NH4+ to Gln conversion process. Equation (2d) describes the evolution of the cell number. The cells can grow on either source in a way that saturates at high concentrations. Specifically, r1 is the maximal growth rate on Gln, and K1 is the concentration of Gln at which half the maximal rate is achieved. The constants r2 and K2 have the same meaning with respect to the alternative energy source R. The CAFs are able to grow when NH4+ is supplied and the concentration of Gln is very low initially. That is why we allow the CAFs to grow on the alternative source. Finally, in Eq. (2d), d1 is the death rate for cells exposed to NH4+, whereas d2 is the natural (or “background”) death rate.

It is likely that R represents the combined effect of multiple factors, such as additional amino acids and metabolites from external and internal sources to provide nitrogen and carbon needed for biosynthesis. We have decided that R which is not accessed in the experiments is present initially at a concentration of 20 mM in all scenarios for the current model. This is for convenience and so that it matches the order of magnitude of all the other concentrations (Gln is supplied at 1, 2 and 4 mM and NH4+ is supplied at 5, 10 and 25 mM, respectively). Note that if R is not replenished, it will tend to zero in the long run. Furthermore, all Gln will be converted to NH4+ and since it is toxic, eventually the cells will die out. This is, however, not observed before 3 days, the duration of our experiments.

The results of the experiments and simulations for the CAFs are shown in Figs. 2 (when NH4+ is supplied) and 3 (when Gln is supplied), respectively. We observe that in all cases the number of CAFs increases, except when NH4+ is delivered at its highest concentration. The 12 parameters are determined by fitting the model output to the experimental data. The numerical values of the parameters for the CAFs together with their 95% confidence intervals are given in Table 1.

Table 1 The fitted numerical values and 95% confidence intervals of the parameters of the model (2a)–(2d) for the CAFs.

Parameter	Value and unit	95% CI	Role	
k1	4.13 × 10−7 (cell × h)−1	[4.03 × 10−7–4.21 × 10−7]	conversion rate of Gln to NH4+	
c1	0.9	[0.85–0.92]	efficacy of NH4+ production	
k2	4.5 × 10−8 (cell × mM × h)−1	[4.06 × 10−8–4.89 × 10−8]	conversion rate of NH4+ to Gln	
c2	0.28	[0.28–0.29]	efficacy of Gln production	
h	1.6 × 10−8 mM × (cell × h)−1	[1.33 × 10−8–1.89 × 10−8]	natural production rate of NH4+	
k3	2.1 × 10−7 (cell × h)−1	[1.83 × 10−7–2.3 × 10−7]	consumption rate of alternative energy	
r1	2.5 × 10−2 h−1	[2.39 × 10−2–2.60 × 10−2]	maximal growth rate on Gln	
K1	0.63 mM	[0.53–0.72]	half-maximal Gln concentration	
r2	3.45 × 10−6 h−1	[3.05 × 10−6–3.84 × 10−6]	maximal growth rate on alternative energy	
K2	7.3 mM	[6.1–8.5]	half-maximal alternative energy concentration	
d1	1.3 × 10−3 (mM × h)−1	[1.27 × 10−3–1.34 × 10−3]	NH4+ induced cell death rate	
d2	7 × 10−5 h−1	[6.33 × 10−5–7.67 × 10−5]	background cell death rate	

Cancer cells grown alone

The model for the cancer cell monoculture is the same as Eqs. (2a)–(2d), with one exception. Since the cancer cells have an increased need for glutamine, they are assumed to be more sensitive to depletion of glutamine and cannot grow on the alternative energy source. Hence we set r2 = 0 from the beginning. The numerical values of the parameters for cancer cells are given in Table 2 and the results of the experiments and simulations are shown in Figs. 4 and 5.

Table 2 The fitted numerical values and 95% confidence intervals of the parameters in the model (2a)–(2d) for the cancer cells.

Parameter	Value and unit	95% CI	Role	
k1	7.5 × 10−8 (cell × h)−1	[7.21 × 10−8–7.89 × 10−8]	conversion rate of Gln to NH4+	
c1	1.9	[1.67–2.18]	efficacy of NH4+ production	
k2	2.4 × 10−8 (cell × mM × h)−1	[2.35 × 10−8–2.41 × 10−8]	conversion rate of NH4+ to Gln	
c2	0.15	[0.14–0.16]	efficacy of Gln production	
h	5.3 × 10−7 mM × (cell × h)−1	[5.1 × 10−7–5.6 × 10−7]	natural production rate of NH4+	
k3	2.6 × 10−10 (cell × h)−1	[1.92 × 10−10–3.37 × 10−10]	consumption rate of alternative energy	
r1	5.6 × 10−2 h−1	[5.4 × 10−2–5.8 × 10−2]	maximal growth rate on Gln	
K1	1.97 mM	[1.93–2.00]	half-maximal Gln concentration	
d1	1.7 × 10−3 (mM × h)−1	[1.6 × 10−3–1.75 × 10−3]	NH4+-induced cell death rate	
d2	1.1 × 10−2 h−1	[9.4 × 10−3–1.3 × 10−2]	background cell death rate	

Figure 4 (A, C and E) The concentrations of NH4+ and Gln when the cancer cells are supplied with NH4+; (B, D and F) the number of live cancer cells.

Figure 5 (A, C and E) The concentrations of NH4+ and Gln when the cancer cells are supplied with Gln; (B, D and F) the number of live cancer cells.

The relative widths of the 95% confidence intervals for all parameters can be used for a sensitivity analysis, as shown in Fig. 6, where all parameters have been normalized to 1. A narrow confidence interval indicates that a parameter is not allowed to vary substantially from its optimal (fitted) value without a considerable increase of the cost function from Eq. (1). On the other hand, a wide confidence interval indicates that the precise value of a parameter is less crucial. While there is not a clearly discernible pattern for both cell types, we see that r1, the growth rate of the cells on Gln is highly sensitive for both. The same holds for d1, the NH4+-induced cell death rate.

Figure 6 The normalized parameter values and their 95% confidence intervals for the CAFs (A) and for the cancer cells (B).

A narrow interval is taken as a sign for a highly sensitive parameter.

The coculture scenario

For the coculture model we denote the number of CAFs by X(t) and the number of cancer cells by Y(t). For all equations of the model we use the corresponding terms from Eqs. (2a)–(2d) for the CAFs, respectively its version for the cancer cells. Then we have the merged coculture model (3a) dAdt=−k1CAFAX+c2CAFk2CAFWRx−k1CCAY+c2CCk2CCWRY,

(3b) dWdt=c1CAFk1CAFAX−k2CAFWRx+c1CCk1CCAY−k2CCWRY+hCAFX+hCCY,

(3c) dRdt=−k2CAFWRX−k2CCWRY−k3CAFRX−k3CCRY,

(3d) dXdt=(r1CAFAK1CAF+A+r2CAFRK2CAF+R−d1CAFW−d2CAF)X,

(3e) dYdt=(r1CCAK1CC+A−d1CCW−d2CC)Y.

Here we have four experimental scenarios, three with with different concentrations of NH4+ (5, 10 and 25 mM, respectively) and one with 4 mM Gln supplied. We use the parameter values from Tables 1 and 2, with only two changes that are given in Table 3. The growth rates of both cell types are allowed to change since in the coculture the it is assumed that the two cell types may affect each other. The results are depicted in Figs. 7–9. We have a good agreement between model predictions and experimental data, except that there is a stronger decline in the total cell number after 48 h. This could be due to depletion of other nutrients such as glucose at that time. In the simulations, the CAFs and cancer cells recover and begin to grow on the newly created Gln. In both experiments and simulations, the ratio between live cancer cells and CAFs remains roughly at 1:1 throughout, except at the highest concentration of NH4+. Our goal is to keep the number of parameter changes relative to the two monoculture scenarios small, as well as the factor by which the parameter is being changed.

Figure 7 (A, C and E) The concentrations of NH4+ and Gln in the coculture scenario; (B, D and F) the number of live CAFs and cancer cells.

This simulation uses the “merged” coculture model (3a)–(3e) with two adjusted parameters from Table 3.

Figure 8 The percentage of live cancer cells in the coculture when treated with (A) 5, (B) 10, and (C) 25 mM NH4+, respectively.

Figure 9 The behavior of the coculture under Gln supply. Shown are the concentrations of Gln and NH4+ (A), the total number of live cells (B), and the percentage of cancer cells (C).

Table 3 The numerical values of the parameters in the coculture model (3a)–(3e).

The third column is the change over their values in Tables 1 and 2, respectively.

Parameter	Value and unit	Ratio	95% CI	
r1CAF	5 × 10−3 h−1	10−1	[4.6 × 10−3–5.4 × 10−3]	
r1CC	6 × 10−2 h−2	1.08	[5.95 × 10−2–6.03 × 10−2]	

The coculture with a stress factor

As we can see in Fig. 7, the merged coculture model (3a)–(3e) performs reasonably well, except for the total cell population at the 72 h data point. In order to remedy this, we introduce a crowding effect or a “stress factor” that is produced by both the CAFs and cancer cells and that kills both cell types in a nonlinear fashion. Thus we add an equation for the stress factor (denoted by L), (4f) dLdt=gXY,L(0)=0,

where g is the rate at which the stress factor is produced by the cells. To Eqs. (3d) and (3e) we add another loss term, quadratic in L, (4d) dXdt=(…−mL2)X,

(4e) dYdt=(…−mL2)Y.

where m is the rate at which the dimensionless stress factor kills the cells (for simplicity, it is the same for both cell types). The stress factor has an increasingly harmful effect at higher elevations. This is necessary to achieve the concave shape of the cell population curve. The additional parameters are given in Table 4 and the simulation results in Fig. 10. However, we have reverted the values of r1CAF and r1CC to their original values from the monoculture scenarios.

Table 4 The numerical values of the additional parameters in the coculture model with the stress factor (3a)–(3c), (4d)–(4f).

Parameter	Value and unit	
g	7 × 10−7 h−1 cell−2	
m	10−11 h−1	

Figure 10 (A, C and E) The concentrations of NH4+ and Gln in the coculture scenario; (B, D and F) the number of live CAFs and cancer cells.

This simulation uses the coculture model with stress factor, (3a)–(3c), (4d)–(4f). The new parameters are listed in Table 4.

Discussion

Cellular metabolism is the process of conversion of chemical compounds from the extracellular environment into storable chemical energy and building blocks for cellular structure (Berndt & Holzhütter, 2011). Metabolism is essential for life, and it is well-known that it is altered in many diseases, including cancer (Markert & Vazques, 2015; Shamsi, Saghafian & Sanati-Nezhad, 2018). The precise way in which the cells’s metabolism is altered often remains insufficiently understood, as well as the implications of the altered metabolism for new cell growth and proliferation patterns. A better understanding of cancer cell metabolism may result in new approaches and targets for treatment of the disease (Enderling & Chaplain, 2014; Yang et al., 2016; Roy & Finley, 2017).

In this article we have performed an experimental-theoretical study of cancer cells, cancer associated fibroblasts and a coculture of both. With some exceptions to which we return below, the simple merged model fits the data rather well. We recall at this point that each data set used to fit a set of parameters consists of all concentrations and live cell numbers for a certain cell type under all treatment scenarios with Gln and NH4+. Recent evidence has pointed out that otherwise “healthy” cells in malignant neoplasms also show altered behavior. Cancer cells often exhibit increased rates to metabolize key nutrients such as glucose and Gln to support macromolecular synthesis. Cellular uptake of Gln is frequently targeted by oncogenic signals (Wise et al., 2008; Dang, 2012; Son et al., 2013).

Toxic NH4+ produced from glutaminolysis is drained by blood vessels and subsequently eliminated via the urea cycle. In addition, cancer cells metabolically remove NH4+, which has just begun to be understood. While some cancer cells are able to convert NH4+ back into amino acids (Spinelli et al., 2017), active involvement of other cell populations plays a critical role. In the breast, normal and cancerous epithelial cells from the luminal lineage express higher levels of GLUL in comparison to basal-lineage cells. This confers a glutamine independence to luminal cells (Kung, Marks & Chi, 2011). The ability of CAFs to recycle metabolic wastes, as we modeled herein, may represent a relatively general mechanism in solid tumors (Yan et al., 2018). This is particularly important for cancer cells carrying mutations in genes related to NH4+ conversion. By engaging CAFs in the tumor microenvironment, which are more tolerant of higher NH4+ concentrations, cancer cells can avoid the high price of cell loss associated with concerting NH4+ by themselves. As reflected by our experimental and mathematical model, the ecosystem comprised of cancer cells and CAFs can maintain itself during short periods of Gln deprivation and NH4+ accumulation by keeping the ratio between the two populations roughly stable. This provides a buffering capacity over hostile metabolic conditions, allowing for sustained tumor growth.

Our mathematical model has produced numerical values for key parameters of our experimental cell lines in Tables 1–4. While a direct comparison is not possible (Collins et al., 1998) show a similar growth behavior of various breast cancer cell lines when supplied with Gln (Fig. 1 in that article). For example, doubling times were found between 20 and 75 h. This is similar to what we observe in our Fig. 5B. These authors also report a rate of Gln disappearance of ∼30 nM per mg cell protein and hour, but this does not translate well into our setting. The authors of Wang et al. (2018), Fig. 1E in that article, report a depletion of glutamine in medium from 2 mM to 0 over 3 days by a population of MCF-7 cells that grows from 103 to 104 during that time. Unfortunately, this is not translated into a parameter value. We anticipate that in the future more and more experimental cell lines will be characterized in a similar fashion, and that eventually a curated database of growth, death and interaction rates will be created, similarly to collections of authenticated cell cultures like ECACC (2020). More available information will be helpful in the process of selecting an experimental cell line in future cancer research (Holliday & Speirs, 2011), in particular as mathematical models become more and more prevalent.

Our mathematical model performs best in the explanation of the monoculture scenarios, while the experimental data from the coculture scenario show a qualitatively different behavior that the mere merger of the monoculture models is not able to explain well. In particular, the total cell number rises initially, and then decreases considerably after 48 h, see Figs. 7B, 7D and 7F. Throughout, the 50/50 composition of the population is essentially maintained, although initially the fraction of cancer cells rises slightly. We propose in an extended model that there is a stress factor produced by the CAFs and cancer cells that increasingly kills both cell types. Future research will have to elucidate the precise reasons for the sharp decline of cell numbers after 48 h, and how a similar population would behave in vivo. It is very likely that since we did not replenish the medium during the cell culturing period due to the need to measure time-dependent consumption and production of glutamine and ammonium, there was a sharp decline in other critical nutrients after 48 h (e.g., glucose and other essential amino acids), which may contribute to the increasing cell death in the coculture scenario.

In addition to explaining the experimental scenarios shown here, our mathematical model can also be used to simulate further possible situations. One such scenario is a “renormalization” treatment of the CAFs by which they return to their normal behavior as fibroblasts. Assuming that the normal fibroblasts do not convert NH4+ to Gln amounts to setting the constant k2CAF in Eqs. (3a) and (3b) to zero. The result of such a simulation is shown in Fig. 11. There, only the cancer cancer cells convert NH4+ to Gln, and the total population decreases faster than when the CAFs convert NH4+ to Gln as well. Other possible scenarios are different ratios of CAFs to cancer cells at time of seeding, different initial amounts of available resources and replenishment of resources at later times. For example, we carried out simulations when one of the two populations dominates initially. This shifts the behavior of the entire population towards that of the monoculture of the dominating population (simulations not shown). It is also possible to replenish the secondary energy source R and thereby to extend the life span of the population (simulations not shown). One feature of our ordinary differential equation model is that it assumes a “well-mixed” environment. A more advanced version of the model would consist of partial differential equations, where the variables also depend on space. In that case one could model the diffusion of the chemical compounds and different initial localizations of the cell types in the coculture. Naturally, this brings with it the need for more parametrization and additional challenges in the numerical simulation.

Figure 11 A comparison of the simulated concentrations of Gln and NH4+ (A) and the total number of live cells (B) of the coculture model (3a)–(3e), for regular CAFs (dashed lines) and for “normalized” fibroblasts (solid lines), where we have set kCAF 2 = 0.

Only the treatment scenario with 5 mM NH4+ is shown.

Conclusion

In future studies, the model presented herein can be amended to include additional cell types found in the tumor microenvironment, such as vascular endothelial cells and immune cells, to understand their role in tumor metabolism and growth and to predict tumor response to therapies such as cytotoxic chemotherapy, metabolic therapy, and immune therapy. One can include a potential interplay between different phenotypes in cancer cells when exposed to therapy (Craig et al., 2019). It would also be interesting to determine the dose effects of various non-cancer cells on both the rate and persistence of tumor growth, and to incorporate not only metabolic interactions but also intercellular crosstalk through growth factors and cytokines, to improve our understanding of the dynamic and highly heterogeneous tumor growth environment. These insights would provide key information towards novel therapeutic strategies targeting the tumor ecosystem as an entity.

Supplemental Information

Supplemental Information 1 Raw data and analysis tools.

Click here for additional data file.

PH thanks the University of California—San Diego for its hospitality during a work visit in September 2017. We thank the editor and three reviewers for a careful reading of the manuscript and valuable comments.

Additional Information and Declarations

Competing Interests

Author Contributions

Data Availability

The authors declare that they have no competing interests.

Peter Hinow analyzed the data, prepared figures and/or tables, authored or reviewed drafts of the paper, and approved the final draft.

Gabriella Pinter analyzed the data, prepared figures and/or tables, authored or reviewed drafts of the paper, and approved the final draft.

Wei Yan performed the experiments, prepared figures and/or tables, and approved the final draft.

Shizhen Emily Wang conceived and designed the experiments, authored or reviewed drafts of the paper, and approved the final draft.

The following information was supplied regarding data availability:

The raw data and the code are available in the Supplemental Files.

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
