# Peer review of "Modeling the bidirectional glutamine/ammonium conversion between cancer cells and cancer-associated fibroblasts"

_PeerJ, doi:10.7717/peerj.10648_

## Round 0.1 · original submission · Major Revisions

Our reviewers are quite positive regarding your manuscript. Several clarifications, additional tests, and sensitivity analyses are, however, required to perfect it. I do not expect any of those requests to be especially burdensome to you. In a personal note, I am afraid that I found Figure 1b to be hard to interpret, and I suggest it be modified (or that additional information is added to the legend for clarification).

Reviewer 1 ·

Basic reporting

The manuscript is well written in general, but in places, the presentation is unclear or lacks sufficient scientific detail. I have included comments and suggestions to the authors.

Experimental design

My one concern is the lack of demonstration of application of the results. In figure 1(a), a 'dotted' arrow is shown as potential therapy targeting the CAF phenotype. The authors should include 1-2 simulations of such a treatment, and what it's effect might be at least in the in vitro case being modeled here.

Validity of the findings

The manuscript meets these criteria.

Additional comments

Fig 1(a) – aspect ratio seems incorrect; image appears stretched horizontally

Fig 1(b) – Please make the legend more descriptive

In Lines 63, 76 etc., the year of the reference should be revisited

Lines 76-82: This feels a little abrupt. You haven’t told us what the monoculture and coculture experiments are. Also, the statement about a decline in cancer cell population feels out of place. In Line 78, ‘our experimental system of cultured breast cancer cells’ is mentioned but this is the first allusion to any experiments herein. A brief overview of the experiments you are presenting here is needed. Also, a more scientific description of the experimental set up is needed, e.g., which breast cancer cell line?

Line 85 – what dynamics are being referred to? Experimental observations? Or model predictions? Is there a figure for reference?

Lines 86 and 88-89 – Where is the evidence to back the claims that CAFs convert all Gln to ammonia but do not convert all ammonia to Gln?

Line 90 – Additions are being made to a model that we have not yet been introduced to

Equations (1)-(4) – if the target audience for this paper is both, mathematicians and biologists, then explain the ‘prime’ notation. Also, it is more standard to denote time derivatives with a ‘dot’.

Lines 95-96 – in describing the dimensionless constants c1 and c2, do you mean, for e.g., that c1 is the number of Ammonia molecules produces for each molecule of Gln consumed?

Equation (3) is confusing – What is the source of this energy? It seems to only decrease. The constant k2 is not introduced in the explanation of the equation, presumably it is the rate of energy consumption as ammonia is converted to Gln? Also, the second term on the right-hand side of (3) is a distinct process of energy consumption, unrelated to Ammonia  Gln conversion? This should be made clear. As I continue to read, I see the explanation of R(t) in the following paragraph but consider reorganizing for clarity.

Line 110 – Are these also the units of concentration for the other chemical species? This should be made clear when introducing the model variables. Please also provide units of time.

Lines 112-113 – From looking at the equations, it is not clear that in the long run, all Gln should be converted to Ammonia. Should this not depend on the choice of parameters k1, k2, c1, c2? Do you prove this claim mathematically? Or show it via numerical simulations?

Line 115 – Figures 2 and 3 are referred to very cursorily. They need to be introduced at appropriate places within the results presentation text preceding this line. So that the reader can look at the experimental or model findings as they are talked about.

Are Figures 2 and 3 the best fits of model simulations to the data? How many parameters are being fitted? Also, the figure legends refer to Figure 2(a) and 2(b) but I see six panels.

Figure 3, bottom panel on the left – Why does the model deviate so strongly from the experiments?

Line 118 – I think the words ‘assumed to be’ are missing, i.e., the sentence should read ‘…they are assumed to be more sensitive to depletion…’. This is a (very reasonable) modeling assumption based on the experimental finding that cancer cells consume more Gln than healthy cells.

Line 123 – What are the different concentrations?

Table 3 – Should r1CC be 5.99x10^-2 h^-1? The notation of using a dot to denote multiplication is confusing with decimals also present. Consider switching to ‘x’.

Reviewer 2 ·

Basic reporting

This is a mostly well-written paper, lacking in details in some sections, that adequately describes the current state of the literature. There could be more focus on models of mono- and co-cultures and their interactions in cancer contexts, including, for example, doi: 10.1038/s41467-017-01174-3, 10.1007/s00109-017-1587-4, and 10.1371/journal.pcbi.1007278. Figure 1A (left panel) can be improved. Instead of marking individual points for triplicates, suggest indicating error bars. Raw data is supplied.

Experimental design

Overall, the experimental design is adequately described. It is unclear to me why numerical simulations were carried out in Mathematica but fitting was done in Matlab, when you would need to implement the model in Matlab to be able to solve it in an optimizer? This shouldn't impact on the results but is curious.

Validity of the findings

A clearer statement of the conclusions could be provided, the discussion reiterates much of what is said as background and context in the introduction. I question how generalizable and valid the results are given the very poor fits to co-culture experiments. Suggest that a sequential fitting procedure could be better adapted and more logical (though details of how each parameter fit was obtained are scarce).

Additional comments

1) Lines 110-111: What is the rationale for choosing an initial concentration of 20nM? Is this due to experimental conditions or prior knowledge? Either should be stated.

2) In all tables, it appears all of the model parameters are fit. Can no prior knowledge be incorporated to reduce the number of parameters? What about using a sequential fitting scheme to fit a reduced number of parameters and then build in others? Perhaps this was done but details were not provided.

3) Lines 118-119: A reference for this statement would be good to provide.

4) Overall, I would suggest using more identifiable variables in the model (this doesn't impact on the mathematics but makes it easier for readers to follow the equations). Also the choice of e as a parameter is not a good one as it looks like an exponential function.

5) Why is no direct interaction between CAF and cancer cells assumed in the model? This seems questionable to me, and I think the lack of this interaction is related to the poor model fits to co-cultures. See, for example, doi: 10.1371/journal.pcbi.1007278. Later on lines 125-126, the authors state: "the growth rates of both cell types are allowed to change since in the co-culture it is assumed that the two cell types may affect each other" but this only modelled indrectly.

6) The discussion can be greatly improved. Lines 152-166 don't belong in a discussion. There is a lack of focus communicating the outcome of the study.

7) Lines 181-190: As stated earlier, the model may be missing important interaction terms that could be leading to these poor fits.

8) Line 234: A better understanding of the interactions of the model could be obtained by varying the initial proportions of cells. What is the rationale for sticking to 1:1 only?

9) "Indicated time points" should be stated explicitly in the methods.

10) Line 245: The objective function should be stated explicitly.

11) To quantify the variance in the normal distributions in the error function, why use bootstrapping replicates while only fitting to the mean data? Why not also incorporate the error from the triplicate experimental observations?

12) Particularly in co-cultures, the fits are not good. How can conclusions be drawn?

·

Basic reporting

The basic reporting aspects of this paper are sound. Figures are clear and readable. Line 76 has a reference to a paper in 2027 that I am assuming is a typo.

I do have some stylistic suggestions. One common style standard is not to begin sentences with variable names or lower case letters, to aid readability. In line 35, "c_1 is. . ." could be rewritten "The variable c_1 is. . ." . In line 105, "That's" may be viewed as too colloquial. The authors may wish to write "That is". The use of the subequations environment in LaTeX can help organize the mathematics. The first system of equations could be labelled (1a) to (1d) rather than (1) to (4) and then referenced collectively as system (1). Similarly with equations (5) to (9).

Experimental design

This is outside of my area of expertise as a mathematician.

Validity of the findings

"Validity" in the sense applied to experimental papers isn't relevant for a work in modeling and simulation like this. The model in this case aims to capture the system of interest and to create an abstract object that can be used to enhance understanding and guide future experimental work. In this sense the work is well on its way to being "valid".

The authors obtain a fair amount of data and use it to fit the parameters for their models. Here the work seems incomplete. It is increasingly de rigueur for ODE models in biology, which tend to have a fair number of fitted parameters, or measured parameters with wide error bars, to have a generalized sensitivity analysis done for the parameter space. Which parameters are the most influential? How much so? What do they represent biologically, and what is the biological significance of them being sensitive or insensitive to perturbations?

---

## Round 0.2 · Minor Revisions

Please address the final suggestions of reviewer #2

Reviewer 1 ·

Basic reporting

no comment

Experimental design

no comment

Validity of the findings

no comment

Additional comments

I am happy with the revised manuscript and can recommend it for publication

Reviewer 2 ·

Basic reporting

In their responses, the authors state that: "the secondary source of chemical energy R is not accessed in the experiments. The choice of 20 mM is to match the order of magnitude of the other, measured, concentrations, which are between 1 and 25 mM." This should be stated clearly in the text.

Experimental design

By “sequential fitting scheme”, I meant taking parameters available in the literature, and fitting individual experiments to a reduced number of variables to reduce the degrees of freedom at each fit.

Validity of the findings

From the conclusion: "Future research will have to elucidate the precise reasons for the sharp decline of cell numbers after 48 h, and how a similar population would behave in vivo." Can the authors suggest some rationale or mechanistic explanation? The "stress" model fits are definitely improved over the original co-culture fits, and the differences between these two models should be informative to the biology underlying the response.

---

## Round 0.3 · accepted · Accept

I am satisfied with your solid responses to the latest reviewer comments. I look forward to seeing your manuscript in print!